# MTSR-MRI: Combined Modality Translation and Super-Resolution of Magnetic Resonance Images

**Avirup Dey**                                                    AVIRUPJU@OUTLOOK.COM
*Jadavpur University, West Bengal, India*

**Mehran Ebrahimi**                                    MEHRAN.EBRAHIMI@ONTARIOTECHU.CA
*Ontario Tech University, Oshawa, Ontario, Canada*

**Editors:** Accepted for publication at MIDL 2023

## Abstract

Magnetic resonance imaging (MRI) is a common non-invasive imaging technique with high soft tissue contrast. Different MRI modalities are used for the diagnosis of various conditions including T1-weighted and T2-weighted MRI. In this paper, we introduce MTSR-MRI, a novel method that can not only upscale low-resolution scans but also translates between the T1-weighted and T2-weighted modalities. This will potentially reduce the scan time or repeat scans by taking low-resolution inputs in one modality and returning plausible high-resolution output in another modality. Due to the ambiguity that persists in image-to-image translation tasks, we consider the distribution of possible outputs in a conditional generative setting. The mapping is distilled in a low-dimensional latent distribution which can be randomly sampled at test time, thus allowing us to generate multiple plausible high-resolution outputs from a given low-resolution input. We validate the proposed method on the BraTS-18 dataset qualitatively and quantitatively using a variety of similarity measures. The implementation of this work will be available at https://github.com/AvirupJU/MTSR-MRI.

**Keywords:** MRI, Super-Resolution, Modality Translation.

## 1. Introduction

High-Resolution MRI produces detailed structural information, helping in clinical diagnosis and accurate quantitative image analysis. However, due to both physical and theoretical limitations, high-resolution (HR) imaging comes at the cost of longer scan times, higher computational costs, and a low signal-to-noise ratio (Plenge et al., 2012). Moreover, for accurate diagnosis, experts may need to obtain HR scans in multiple modalities such as T1, T2, and FLAIR. In this paper, we propose a method that can - (i) Perform Image Super-Resolution (SR) i.e., restore an HR image from a single low-resolution (LR) input, and (ii) Perform Modality Translation (MT) i.e., translate an MR image across multiple modalities that have different contrast settings.

Image Super-Resolution (SR) is a challenging problem in the area of medical imaging and computer vision since there can be multiple HR images corresponding to a single LR image (Tanno et al., 2017). The ambiguity is particularly pronounced for high upscaling factors, for which texture details in the reconstructed SR images are typically absent. While filtering approaches, e.g. linear, bicubic or Lanczos (Duchon, 1979) filtering, can be fast to obtain, they fail to reconstruct fine image details and yield overly smooth textures. In recent years, several deep learning-based methods have been proposed (Ledig et al., 2017;

Wang et al., 2018, 2021) that have shown promising results in upsampling natural images[1]. However, these architectures often require large datasets and high computation power for training. Thus, we need to optimize the architecture and training process to leverage the inherent sparsity and redundancy in MR imaging data which would result in faster training and inference. Our second objective is image modality translation. Several promising

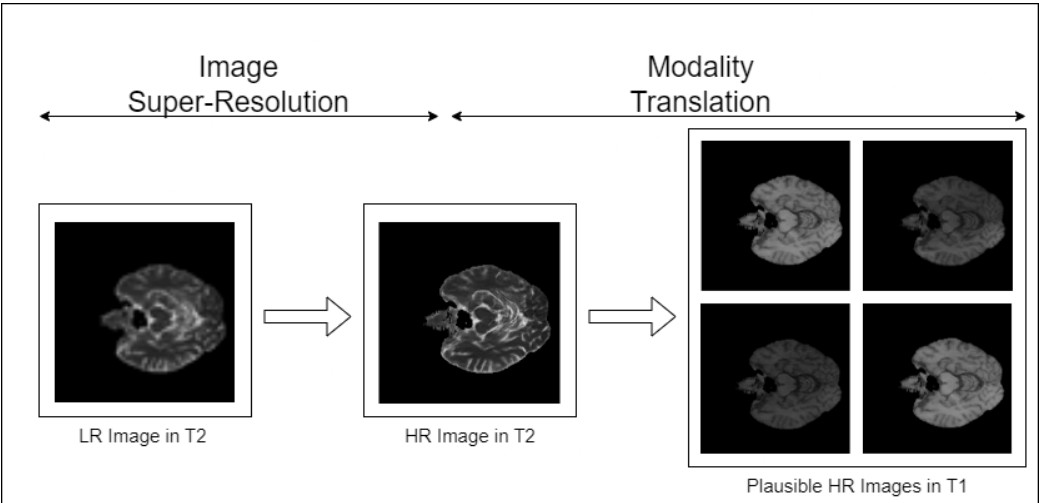

Figure 1: **Overview of the Method**: First, we reconstruct an HR image from a given LR input in Modality A (T2 in this case), and then, we use this image to generate plausible HR outputs in Modality B (T1 in this case). Instead of using a bijective mapping between modalities, we model a distribution of potential outputs, that yields an array of images corresponding to a range of acquisition parameters.

methods have been proposed with the emergence of generative adversarial networks (GANs) (Goodfellow et al., 2014). The most notable of them[2] are based on Pix-2-Pix (Isola et al., 2017), for paired image translation, and CycleGAN (Zhu et al., 2017a) for unpaired images. Nevertheless, image-to-image translation in many cases is a poorly defined task as an image in one domain may be mapped to multiple images in another domain. Thus, to account for all the possible variations, we need to learn a distribution of plausible outputs. In this work, we propose MTSR-MRI, a novel framework, that jointly addresses the aforementioned problems, see Figure 1. First, we employ an optimized super-resolution model that takes in an LR input in one modality and upsamples it to produce an HR image. Then, this intermediate image is fed to an image-to-image translation model that maps it to a set of plausible images in another modality, using variational inference. In summary, our contributions are as follows.

- We formulate a novel method to jointly solve the problem of resolution enhancement and modality translation of MRI. The method could potentially reduce scan times.

---

1. Previous research in MRI Super-Resolution has been cited in Appendix A.1.
2. Please refer to Appendix A.2. for previous works.

- We optimize the super-resolution model by modifying its architecture and training process that would allow us to leverage the sparsity in MR images.

- We perform extensive quantitative analysis of the proposed method on the BraTS-18 dataset (Menze et al., 2014; Bakas et al., 2017, 2018) using various similarity metrics to demonstrate its efficacy.

## 2. Proposed Method

### 2.1. Overview and Notations

Our aim is to model a function $\mathcal{F}$ that maps an LR image in Modality A, i.e, $(I_{LR}^A)$, to an HR image in Modality B, i.e., $(I_{HR}^B)$. Mathematically, we represent this task as

$$I_{LR}^A \xrightarrow{\mathcal{F}} I_{HR}^B. \tag{1}$$

Now, $\mathcal{F}$ can be represented as a composition of two functions, $\mathcal{F}_1$ and $\mathcal{F}_2$ such that

$$I_{LR}^A \xrightarrow{\mathcal{F}_1} I_{HR}^A, \tag{2}$$

$$I_{HR}^A \xrightarrow{\mathcal{F}_2} I_{HR}^B. \tag{3}$$

**Equation (2) is an SISR (Single Image Super-Resolution) problem**. We know, the LR image is just a downsampled version of the HR image. Hence,

$$I_{LR}^A = \mathcal{D}_\downarrow(I_{HR}^A) + \mathcal{N}. \tag{4}$$

Here, $\mathcal{D}_\downarrow$ denotes a downsampling kernel of some known integer factor, and $\mathcal{N}$ denotes a combination of sensor noises and imaging artefacts, modelled as an additive expression. Thus, $\mathcal{F}_1$ is an inverse mapping from $I_{LR}^A$ to $I_{HR}^A$. We learn this function using a deep learning-based super-resolution model described in Section 2.2.

**Equation (3) is an image-to-image translation problem**, where the aim is to obtain a mapping between Modality A and Modality B. It is important to note that there may be multiple plausible $I_{HR}^B$ corresponding to an image $I_{HR}^A$, due to variations in scanning parameters or design specifications of different scanners. Hence, we need to learn a distribution of latent vectors that might correspond to potential images in Modality B, given an image in Modality A. Thus $\mathcal{F}_2$ is a function that returns a plausible image in Modality B, given an input of Modality A and a sample from the latent distribution of Modality B,

$$I_{HR}^B = \mathcal{F}_2( I_{HR}^A, z_B ). \tag{5}$$

We learn this function and the latent distribution using a VAE-GAN-based framework (Larsen et al., 2016), described in Section 2.3.

### 2.2. MRI Super-Resolution Model

We build the proposed super-resolution (SR) model based on ESRGAN (Wang et al., 2018). The generator uses RRDB (residual in residual dense blocks), which has been proven to be effective for (Peak signal-to-noise ratio) PSNR-oriented tasks such as deblurring (Nah

et al., 2017) and upsampling (Lim et al., 2017). The dense connections within the block help in preserving the local structure and low-level details of the image. Nevertheless, the generator architecture is bulky and adversarial training with a relativistic discriminator (Jolicoeur-Martineau, 2018) increases the computational cost. To mitigate these performance bottlenecks, we leverage the sparsity in MR images. We note that most of the areas in these images are zero or constant (also referred to as 'empty') which implies most of the layers in the generator 'learn' nothing. Hence, we empirically select a subset of the ESR-GAN generator *(Refer to Appendix B. for implementation details.)* for our usage. To speed up the training even further, we replace the relativistic discriminator with a PatchGAN discriminator (Li and Wand, 2016).

### 2.2.1. Loss Functions

Besides the adversarial loss we use a VGG-based feature loss (Johnson et al., 2016) to optimize the overall perceptual quality and $L_1$ loss for the reconstruction of pixel-level details. Thus, the overall objective becomes

$$\mathcal{L}_G = \mathcal{L}_{GAN} + \lambda_1 \mathcal{L}_{VGG} + \lambda_2 L_1. \tag{6}$$

Here, $\lambda_1$ and $\lambda_2$ are hyper-parameters set to 200 and 2, respectively.

### 2.2.2. Training Details

We train the model for 100 epochs with a batch-size of 12 and a learning rate of $2 \times 10^{-4}$. We also pre-train the generator using $L_1$ loss for 10 epochs to obtain suitable results. The reasons are that (a) it can avoid undesired local optima for the generator; (b) after pre-training, the discriminator receives super-resolved images instead of fake ones (black or noisy images) at the very beginning, which helps it to focus more on texture discrimination.

### 2.2.3. Dataset Preparation

Following SRGAN (Ledig et al., 2017), all experiments are performed with a scaling factor of x4 between LR and HR images of 2 dimensions. We employ a second-order degradation method (Wang et al., 2021) to obtain realistic LR samples, as shown in Figure 2.

## 2.3. MRI Modality Translation Model

We intend to model the distribution of the target modality such that decoding sampled vectors from it would produce realistic and diverse outputs. The VAE-GAN approach (Larsen et al., 2016) is to first encode the target image into the latent space, giving the generator a noisy "peek" into the desired output. Using this, along with the input image, the generator should be able to reconstruct the specific output image. To ensure that random sampling can be used during inference time, the latent distribution is regularized using KL-divergence to be close to a standard normal distribution. Another approach (Donahue et al., 2016; Dumoulin et al., 2016) is to first provide a randomly drawn latent vector to the generator. In this case, the produced output may not necessarily look like the ground truth image, but it should look realistic. An encoder then attempts to recover the latent vector from the output image. We build our modality translation model (MT) based on the

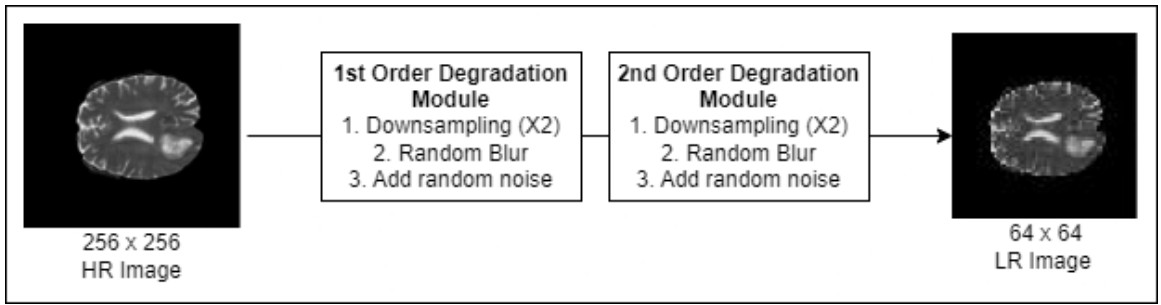

Figure 2: **LR Image Generation**: Each degradation block reduces the spatial dimensions by ×2, convolves it with a blur filter (random choice among Gaussian, isotropic, anisotropic, etc), and finally adds noise (random choice among Gaussian, Poisson, etc ), giving us an LR image with dimension 64 × 64.

work of Zhu *et. al.* (Zhu et al., 2017b), a hybrid of the aforementioned methods. Figure 3 outlines the training process of the proposed model.

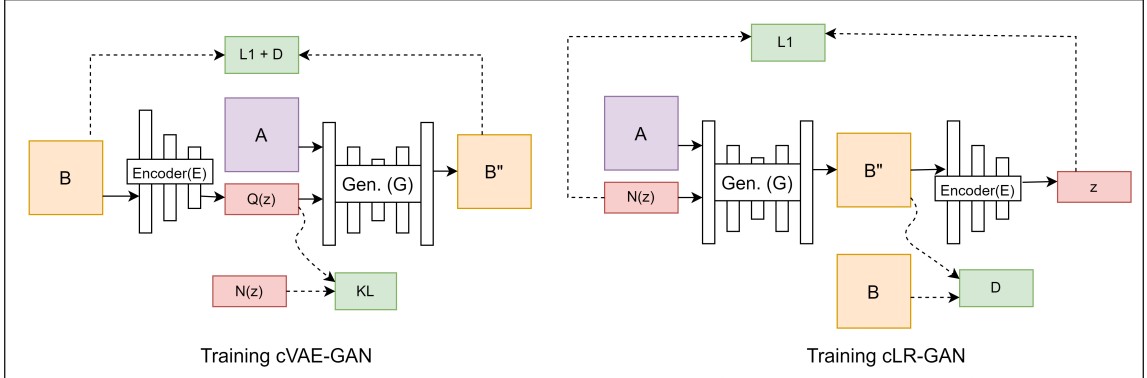

Figure 3: **Training the MT model**: On the left, we have the cVAE-GAN that starts from a target image B and encodes it into the latent space. The generator then attempts to reconstruct B using the input image A along with a sampled z. On the right, we have the cLR-GAN that samples a latent code from a normal distribution, uses it to map input A to output B, and then tries to reconstruct the latent code from the output. We optimize the proposed model using both cycles. At inference, the generator G uses the input A and a sampled latent vector z to construct the target image B.

At inference, the generator takes in an input of modality A ($I_A$) and a latent vector, $z_B$ from the learned distribution and returns a plausible image of modality B ($I_B$).

### 2.3.1. Loss Functions

We combine the objectives of VAE-GAN (Larsen et al., 2016) and cLR-GAN (Donahue et al., 2016) to utilize both the cycles ($B \rightarrow z \rightarrow \hat{B}$ and $z \rightarrow \hat{B} \rightarrow \hat{z}$). The overall objective is

$$\mathcal{L}_{total} = \mathcal{L}_{GAN}^{VAE} + \lambda\mathcal{L}_1^{VAE} + \mathcal{L}_{GAN} + \lambda_{latent}\mathcal{L}_1^{latent} + \lambda_{KL}\mathcal{L}_{KL}, \tag{7}$$

where the hyper-parameters $\lambda$, $\lambda_{latent}$, and $\lambda_{KL}$ are tunable hyper-parameters.

### 2.3.2. Training Details

We train the proposed model for 100 epochs using a batch size of 1 and an initial learning rate of 0.0002. The hyper-parameters $\lambda$, $\lambda_{latent}$, and $\lambda_{KL}$ are set to 10, 0.5 and 0.01 respectively. Additionally, we choose a least-squares loss instead of a cross-entropy loss for our GAN objective as suggested by Mao *et. al.* (Mao et al., 2017).

## 3. Experiments

### 3.1. Qualitative Results

In Figure 4 we have compared the reconstructions by the proposed model to traditional upsampling methods of Bilinear and Bicubic interpolation as well as the state-of-the-art deep learning model, ESRGAN. We note that ESRGAN preserves contrast and low-level texture more accurately compared to the proposed model. This can be attributed to the fact that the proposed model only uses a subset of the ESRGAN generator, and we have omitted the relativistic discriminator for speeding up the training process, which led to random smoothening artefacts at convoluted edges. Figure 5 shows sample results for the T2-to-T1 translation task. We note that the proposed model produces results that are diverse as well as perceptually realistic.

### 3.2. Quantitative Results

We sampled 150 images from the BraTS-2018 dataset (Menze et al., 2014; Bakas et al., 2017, 2018) for all evaluation purposes. Table 1 shows a quantitative comparison between various resolution enhancement methods using similarity measures like PSNR and SSIM, as well as inference time on the test set. We note that the proposed model scores competitively against state-of-the-art methods such as ESRGAN while being twice as fast. Table 2 compares various image-to-image translation models for T1-to-T2 and T2-to-T1 mapping using the LPIPS distance, for realism and diversity[3]. Although Pix2Pix produces realistic translations, injecting noise into the latent vector does not produce much variation in the output, thus leading to low diversity score. This confirms the findings of Isola *et. al.* (Isola et al., 2017) that noise addition produces very marginal deviations in the output. Similarly, CycleGAN produces fixed point outputs that do not vary much with respect to the injected noise. The proposed model, on the other hand, generates translations that are perceptually realistic as well as diverse and hence, can account for variations in scanning parameters.

---

3. Please refer to Appendix C for details

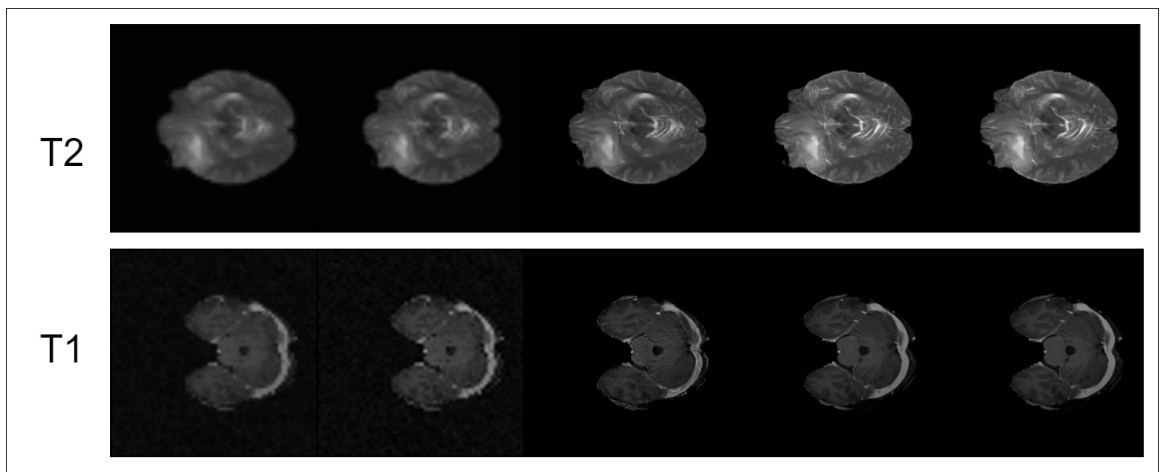

Figure 4: **SR on T1 and T2 Images**: [Left to Right] Bilinear interpolation, Bicubic interpolation, Proposed, ESRGAN, Ground Truth. Although the proposed method outperforms traditional upsampling methods, it falls behind ESRGAN in restoring low-level details. (Zoom in for a closer view)

Table 1: Quantitative results on the BraTS-18 dataset.
The best performance is highlighted in bold, and the second best is underlined.

| Method | T1 Modality | | T2 Modality | | |
|---|---|---|---|---|---|
| | SSIM ↑ | PSNR ↑ | SSIM ↑ | PSNR ↑ | Inference Time (s) ↓ |
| Bilinear | 0.69 | 31.64 | 0.65 | 32.71 | **1.384** |
| Bicubic | 0.75 | 32.01 | 0.77 | 32.93 | **1.396** |
| SRGAN | 0.87 | 37.33 | 0.83 | 35.74 | 22.998 |
| ESRGAN | **0.92** | **38.02** | **0.93** | **38.10** | 23.211 |
| **Proposed** | 0.90 | 37.65 | 0.89 | 36.64 | 11.287 |

Table 2: Quantitative results on the BraTS-18 dataset for Modality Translation.
The best performance is highlighted in bold, and the second best is underlined.

| Method | T1-to-T2 | | T2-to-T1 | |
|---|---|---|---|---|
| | Realism ↓ | Diversity ↑ | Realism ↓ | Diversity ↑ |
| Pix2Pix (w/ noise injection) | **0.063** | 0.009 | **0.071** | 0.011 |
| CycleGAN (w/ noise injection) | 0.162 | 0.022 | 0.149 | 0.023 |
| Proposed | 0.159 | **0.110** | 0.166 | **0.102** |

## 4. Conclusion

In this paper, we proposed a novel deep learning-based framework to jointly address image super-resolution and image modality translation in MRI.

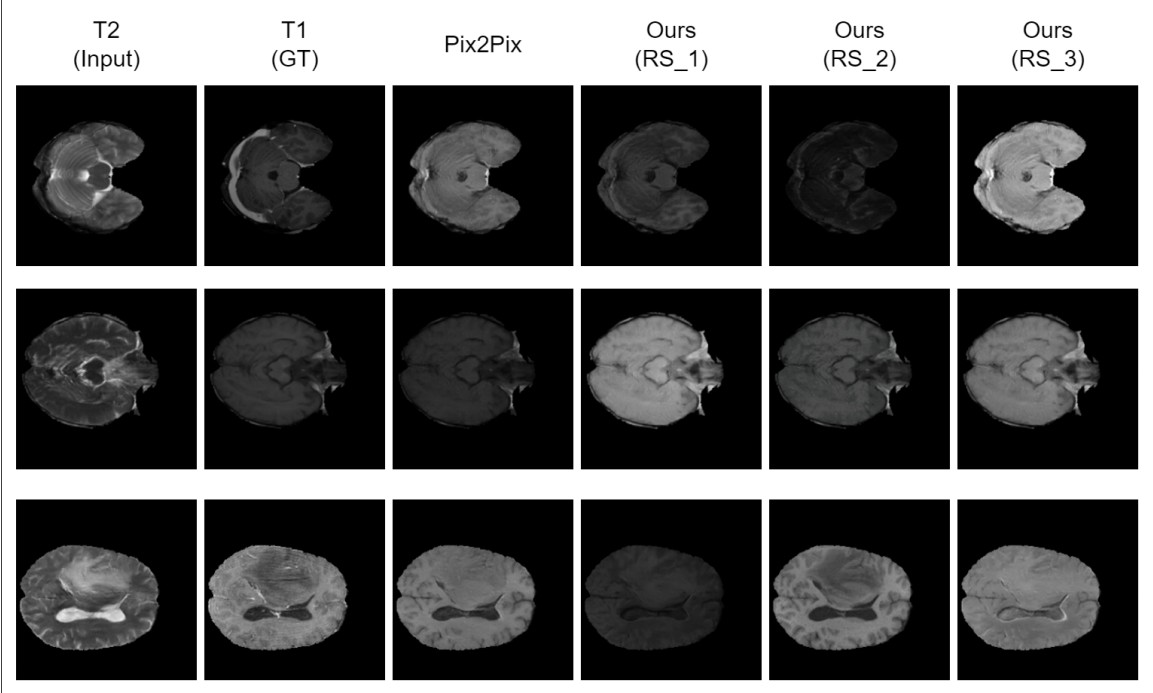

Figure 5: **MT from T2 to T1**: Random samples (RS) of T1, generated by the proposed modality translation (MT) model, are not only perceptually similar to the ground truth but also diverse, hence, might correspond to a range of T1 scanning parameters. It should be noted that the sampling is random in nature. (Zoom in for a closer view.)

In a clinical setting, the radiologists could inspect translated samples (T1-to-T2 to T2-to-T1, as per requirements) to reach a preliminary decision. The proposed method can potentially aid their decision-making process. Computational methods such as the proposed scheme, primarily aim to avoid multiple scans per patient and reduce overall scan times. We do not claim perfect precision, nor recommend making a final inference based on our model's output. All Computer-aided diagnosis (CAD) systems require expert handling and in this work, we have proposed a method as a baseline to potentially reduce some workload.

For generating the LR images, we have used a series of degradations that includes blurring, down-sampling, and adding noise. Although different from the actual mechanism of obtaining LR MRI (e.g., using raw data and sampling in K-space), we believe the proposed forward model provides measurements using an approximation of the true kernel that can be generalized in the context of raw data, if needed. Designing/testing SISR models on actual LR-HR pairs in the raw K-space format would yield more accurate results, consistent with MR physics. Nevertheless, the employed degradation method approximates quite well - tissue structures and intricate contrast details are indeed lost consistent with the physical model, although not precisely using the same degradation kernel.

Our intention in this work was to inspire research in addressing the two problems of resolution enhancement and modality translation in a combined/joint fashion. One can always modify the dataset and make architectural tweaks to improve the performance by working directly with the raw data, if available. Our aim here was to generate results inferred directly from the original inputs for which we could measure and assess standard image quality measures. The proposed super-resolution model not only scored at par with state-of-the-art SISR methods but also outperformed them in terms of training efficiency and inference time. Furthermore, the proposed modality translation model accounted for variations in scanning parameters for T1 and T2 modalities to produce diverse and realistic images. To our knowledge, this is the first work that addresses SR and MT in a combined way using deep learning. Nevertheless, we acknowledge the limitations of the proposed method.

Random sampling must be interpreted with care, especially in the clinical setting. Future works can explore this domain and find novel generative methods that allow more control over the model outputs.

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

## Acknowledgments

This work was supported in part by the Natural Sciences and Engineering Research Council of Canada (NSERC) in the form of a Discovery Development Grant for Mehran Ebrahimi. Avirup Dey was supported by a Mitacs Globalink Research Internship at Ontario Tech University.

## Appendix A. Previous Works

### A.1. Image Super-Resolution in MRI

The spatial resolution of MR images may be degraded due to the constraints such as scan time, theoretical limitations of the process, and hardware configurations (Ebrahimi and Bohun, 2021; Ebrahimi and Martel, 2009; Abedjooy and Ebrahimi, 2022; Rashid et al., 2022; Ebrahimi et al., 2009; Ebrahimi and Vrscay, 2007; Nazeri et al., 2019; Ebrahimi and Vrscay, 2008; Narang et al., 2022; Bouffard et al., 2023).Besides the traditional upsampling methods, several deep learning-based methods have been proposed for MRI SR. Zeng *et. al.* (Zeng et al., 2018) proposed a method estimating single-contrast and multi-contrast MRI images simultaneously. Shi *et. al.* (Shi et al., 2018) introduced a local residual block and global residual network to extend SRCNN to solve a 2D MRI SR problem. Zhao *et. al.* (Zhao et al., 2019) proposed an extended a SRCNN architecture for 2D MRI brain images. The network consists of three main sub-networks: feature extraction sub-network, non-linear mapping sub-network, and reconstruction. The general idea is to split features at one channel into two branches so that the features at the same channel can be processed differently. Liu *et. al.* introduced a multi-scale fusion convolution network (MFCN) that has several multi-scale fusion units (MFU) to estimate restoration filters at specified scales. A set of MFU helps to reconstruct super-resolution images with different scale features. Oktay *et. al.* (Oktay et al., 2017) proposed a multi-task network that could perform SR and segmentation that used SRGAN (Ledig et al., 2017) for SR. GAN-CIRCLE (Lyu et al., 2019) used cycle consistency to ensure structural similarity between LR-HR pairs.

### A.2. Modality Translation in MRI

Obtaining MRI scans of multiple modalities is crucial for diagnosis as they provide complementary information. However, data acquisition might be limited due to time constraints, scanning costs, and the patient's comfort. Hence image modality translation is crucial for this domain. Yang *et. al.* (Yang et al., 2018) proposed a cGAN-based method similar to Pix-2-Pix (Isola et al., 2017) for translating between T1 and T2 modalities. Later, they also appended an FCN-based module that performed segmentation of images from both modalities (Yang et al., 2020). Dar *et. al.* (Dar et al., 2019) employed cycle consistency for image translation thus enabling their model to work with unpaired MRI data. Based on the said works, Yu *et. al.* (Yu et al., 2019) introduced Edge-Aware GANs that incorporated edge maps while training, leading to better preservation of low-level details at the edges. Huang *et. al.* (Huang et al., 2017) was the first to jointly address super-resolution and modality translation in MRI. They proposed a weakly-supervised joint convolutional sparse coding framework that requires only a few registered multimodal image pairs as the training set and achieved impressive results on multiple brain MRI datasets.

## Appendix B. Architectural Details of the proposed SR Model

The ESRGAN generator has 23 RRDB blocks. The dense connections as well as the overall depth of the architecture enables it to learn complex natural images (Timofte et al., 2017, 2018) but the inherent "sparsity" of the MR images, allows us to compress the architecture in order to reduce computation. Table 3. shows the results of the ablations we performed

Table 3: Model Performance vs Speed for different variants of ESRGAN that were obtained by pruning out specific RRDB blocks.

| ESRGAN Variant | Removed RRDB | No. of RRDB in model | PSNR | SSIM | Inference Time (s) |
|---|---|---|---|---|---|
| v0 | None | 23 | **38.10** | **0.93** | 23.211 |
| v1 | (1,2,22,23) | 21 | 37.92 | 0.93 | 20.115 |
| v2 | [1-4], [20-23] | 15 | 35.88 | 0.89 | 15.003 |
| v3 | 1-11 | 12 | 34.03 | 0.84 | 11.268 |
| v4 | 12-23 | 12 | 35.22 | 0.81 | 11.274 |
| **Proposed** | All Even Blocks | 12 | 36.64 | 0.89 | **11.287** |

on the architecture to find a configuration that was optimized for quality and speed. We initialize the architectures with weights of a model trained on the DIV2K dataset and fine-tune them with the hyper-parameters mentioned in Section 2.2.2. We note that compressing the model by pruning out every alternate layer does not degrade the reconstruction quality significantly despite having almost half the number of parameters. The shallower layers help in the restoration of low-level details and the deeper layers help in improving texture and overall perceptual quality. Figure 6 shows the outline of the proposed generator architecture.

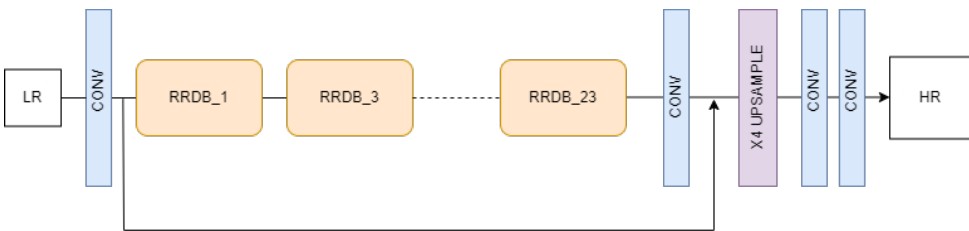

Figure 6: **Generator Architecture**: We select every alternate block from the ESRGAN, pre-trained on natural outdoor images. This helps in reducing computation without significantly degrading the quality of the reconstruction.

## Appendix C. Measuring Realism and Diversity of GAN Outputs

Following Zhu *et. al.* (Zhu et al., 2017b), we use LPIPS distance to evaluate the proposed modality translation model. This measure has been shown to match human perception better than FID, CAS, etc (Borji, 2022). A low LPIPS score means that images are perceptually similar. Conversely, a higher score would indicate greater dissimilarity between images. Hence, LPIPS can be used as a measure for realism as well as diversity in synthetic images. To measure diversity, we first produce 5 translations by randomly sampling 5 latent vectors (z), for each image. We then compute LPIPS distance between consecutive pairs to get 4 paired distances. Taking a mean of these distances across the entire test set gives us the measure for diversity. A higher value indicates greater diversity.

