# OpenReview forum: "MTSR-MRI: Combined Modality Translation and Super-Resolution of Magnetic Resonance Images"
_MIDL.io/2023/Conference — MIDL 2023 Poster_

### Official Review · Reviewer_BLxN · 2023-02-03

**Confidence:** 4
**Preliminary Rating:** 2
**Recommendation:** Poster

**Summary:**

The paper introduces a new tool called MTSR-MRI, which enables a joint modality translation and the generation of super resolution MRI images. The validation step uses the BraTS-18 data set and provides a wide series of experimental runs that were completed.

The clarity of the submission adequate.






**Strengths:**

The generation of high resolution multi-modal image pairs from low res images is very appealing, esp with high cost MRI.

Efficient and fast computations of these high resolution images is highly desirable.

**Weaknesses:**

The frameworks on which the pipeline is built seem to be well established (~5 yrs). While the frameworks on which the pipeline is built seem to be well established (~5 yrs), the combination of the MT and SR components is not and should be novel. However, how different is the proposed solution from Shu2017? My current impression is that it is the same.

The figures should be better utilized. Better quality image data should be displayed to help the reader and also higher resolution plots.

I found that some of the claims are a bit ambitious. Clinical images are not going to be generated from a lower resolution version for medical diagnosis. However the proposed tool could certainly be useful for generating missing images for longitudinal cohorts, population studies and tool development.

Maybe needs a bit more of follow-up and positioning in the literature. Where is it that the outputs of this tool is intended to be used? Is that an intermediate step or is it the final result?

The set of possible / plausibility HR outcomes is very wide. The authors claim that the distribution of plausible generated outputs has to be wide. I need a bit more explanation here. Is it that the tool needs to be able to generate a wide variety of solutions (as a capability)? What is to be done with all those? How will this be narrowed down to a validated set in the future?

Maybe needs a bit more of follow-up and positioning in the literature. Where is it that the outputs of this tool is intended to be used? Is that an intermediate step or is it the final result?

The proposed tool is the 2nd best, performing consistently. ESRGAN is the top performer. Are there any outcomes that were statistically significant? The tool trains faster but its performance is worse than other state-of-the-art work. What is the goal performance that the readers / authors should be satisfied with?

**Deanonymize Review:**

no

**Detailed Comments:**

The following acronyms were not defined in the paper: PSNR, SSIM, LPIPS SSIR

The medical images in the figures are very poorly visible. The authors should work on improving these as currently the anatomy details cannot be appreciate. (Fig1, Fig4) Use either better CNR or white background. Currently it is very hard to appreciate the input images.

Also minimize the background and zoom in on the important part of the image.

Fig 4: Can you point to particular areas and guide the reader? Authors should use arrows, circles around the anatomy in question.
          This also confuses me. Is that really the goal to reconstruct a very bad quality scan? Shouldn't it be the goal to reconstruct the non-
           motion-corrupted version from information learned from the other data?

Fig 5: This is all in 2D data correct?

Fig 6:  higher resolution figure would be better

Nothing was said about the BRATS data set. That would help the reader with understanding what is in the data set.

**Paper Type:**

both

**Questions To Address In The Rebuttal:**


Maybe it is expected from them to ignore noise and recover another modality without noise? I understand point about multiple modalities, but then how many will be generated as there could be infinitely many.

Can you describe an eventual application area / future research?  Without such information that it is hard to interpret whether the current performance level is sufficient.

Can the authors clarify whether the proposed formulation is scalable to 3D?

---

### Official Review · Reviewer_jYUo · 2023-02-05

**Confidence:** 4
**Preliminary Rating:** 5
**Recommendation:** Oral

**Summary:**

This paper addresses the typically long scan  time in MRI by suggesting a mapping technique from low res image to hi8gh res while also translating between T1-weighted and T2-weighted contrasts (hence again reducing time if potentially only one of these needs acquiring).
The method is evaluated quite extensively on the Brats-18 dataset.

**Strengths:**

This is a lovely paper, with excellent methods and intro parts detailing in good schemata and overviews the processing steps applied and the innovations suggested. Evaluation on the BRATS-18 dataset allows to appreciate the functionality.
The combination of SR and MT is great and makes a lot of sense - it opens a lot of avenues for future developments and improvements as well and is thus a paper which potentially leads to a lot of discussion and interest.

**Weaknesses:**

Not much discussion and mentioning of limitations - there are obvious areas of improvement such as a better evaluation of the generated contrast (eg how well do generated T1 images perform in different setting for which these are often used such as parcellations, radiological measures, detection of blood etc,,).
In addition, the images could be quite a lot improved (see below as well).

**Deanonymize Review:**

no

**Paper Type:**

methodological development

**Questions To Address In The Rebuttal:**

The images all need a bit of tweaking (these with actual images not the schemata these are lovely!) and adding a bit of brightness I think! This is something which could be addressed in the rebuttal easily. (Figure 1 and 4 are examples for this)

---

### Official Review · Reviewer_sV6K · 2023-02-07

**Confidence:** 5
**Preliminary Rating:** 2

**Summary:**

In this paper a multi-modality (MM) super-resolution (SR) for magnetic resonance images (MRI) is presented. High resolution MRIs from BraTS-18 dataset are engineered to a low resolution format and a deep learning method is tasked to reconstruct a high resolution MRI (SR task) of a different sequence (MM task).

**Strengths:**

The idea is really interesting. Ability to generate HR MRI sequences, starting from LR MRI of different sequences is really important for multiple reason which could include 1) to improve patient comfort (shorted scan time), 2) reduce health care cost (fewer sequences to be scanned), 3) enhance clinical assessment (multiple information about patient health).

**Weaknesses:**

LR MRIs are the result of a coarser acquisition, fewer slices are acquired, larger spacing among slices, lower in plane resolution. These are all conducted in k-space, and as a result the resulting MRIs might not be as sharp as HR, but most importantly might not include details in the tissue. This is in principle different from the blurring done here, which seems to be more suitable to natural images. Because of this it is hard to celebrate merits of the work.

**Deanonymize Review:**

no

**Detailed Comments:**

In addition to what referred to the weakness section, the qualitative analysis does not include a clinical assessment. Methods utilized for comparison do not really include methods related to MRI recon or super resolution, and there are a few out there. Also a comprehensive related work section is missing. The architecture and the loss function have multiple components, are they all necessary? an ablation study could have been useful too.

**Paper Type:**

validation/application paper

**Questions To Address In The Rebuttal:**

The rebuttal could be an opportunity to clarify if comments above are the results of a big misunderstanding from my side, and provide additional evidence that the method would be indeed useful in clinical practice.

---

### Official Review · Reviewer_k3h5 · 2023-02-07

**Confidence:** 5
**Preliminary Rating:** 3
**Recommendation:** Poster

**Summary:**

The paper introduced a framework for MR images that can conduct super-resolution tasks for low-resolution scans and translate between the T1-weighted and T2-weighted modalities. The authors claimed that the proposed method achieves less training and inference time when compared with other baselines.

**Strengths:**

The motivations of the work seem to be great. The authors aimed to propose a framework that can perform both super-resolution and image translation and optimize the model training and inference efficiency.

**Weaknesses:**

Both super-resolution and image synthesis pipelines are relatively not recent.  The diffusion model achieves decent super-resolution results these days. The authors should at least discuss this in the related work.

As an application paper, the paper needs more detail about how the brats-18 data get pre-processed (i.e., normalization, training with 3D or 2D, hardware detail), which could bring trouble to the reproducibility of the work because brats-18 is a public dataset.

The image representation of this work needs to be clarified; T1 and T2 representations need to be more accurate. Please check the detailed comments.

**Deanonymize Review:**

no

**Detailed Comments:**

- Figure 1, the top right of 'plausible HR images in T1' doesn't seem to be T1 but more like T2. Please check. Similarly, Figure 4's T2 and the second row of Figure 5 don't look like T2 but more like T1.
- The proposed method indeed conserves 50% inference time. However, it is hard to know if it is good enough. Maybe adding more use case that needs real-time scanning can help the story.

**Paper Type:**

validation/application paper

**Questions To Address In The Rebuttal:**

For Figure 1, 'Instead of using a bijective mapping between modalities, we model a distribution of potential outputs that yield an array of images corresponding to a range of acquisition parameters.' it is unclear how the plausible HR images in T1 are merged to generate the final synthesis result.

If the image looks real or the image is useful for downstream analysis, could the authors explain why diversity matters? It is a known, challenging issue in 3D radiology image synthesis, so adding a downstream analysis can improve the proof-of-the concept.

It is unclear to train pix2pix and cycleGAN why injection noise is needed. Did the authors add this step for data augmentation?

---

### Meta-Review · Area_Chair_BGuD · 2023-02-22

**Recommendation:** Accept (Poster)
**Confidence:** 4

**Metareview:**

Based on the reviewer comments and my own reading of the paper,
Strengts:
1. The generation of high resolution multi-modal MR images from a low resolution MRI scan of a single modality  may have practical use, given the high cost of MRI.
2. The main contribution is the joint framework for performing super-resolution and modality translation (image translation problem) together, which is claimed to allow faster inference.

Weaknesses:
1. The proposed model isn’t the best in most respects — it underperforms ESRGAN in the super resolution task and pix2pix in the translation task (albeit with larger inference time and lower diversity, respectively), and it is not clear how important inference time and diversity are, in this particular context
2. Does not consider more recent diffusion model based methods which have proven quite good in related tasks.
3. The evaluation is not very comprehensive — no radiologists’ input/evaluation of the generated image quality, no discussions about ‘hallucinations”.
4. Current figures are very low quality but the authors have promised to rectify that in the camera-ready version

I think the paper is a pilot study and while it has many weaknesses, it might be of interest in the community.